computational chemistry

comparative, analysis, mechanisms, DFT, ammonia synthesis, reaction mechanism

**Authors for correspondence:**
Constantinos D. Zeinalipour-Yazdi
e-mail: c.zeinalipour-yazdi@uel.ac.uk
C. Richard A. Catlow
e-mail: c.r.a.catlow@ucl.ac.uk

This article has been edited by the Royal Society of Chemistry, including the commissioning, peer review process and editorial aspects up to the point of acceptance.

Catalysis for a Sustainable Future.

# A comparative analysis of the mechanisms of ammonia synthesis on various catalysts using density functional theory

Constantinos D. Zeinalipour-Yazdi[1,2],

Justin S. J. Hargreaves[3], Said Laassiri[4] and

C. Richard A. Catlow[5,6]

[1]School of Health, Sport and Bioscience, University of East London, Stratford Campus, Water Lane, London E15 4LZ, UK
[2]Department of Natural Sciences, Middlesex University, Hendon Campus, The Burroughs, London NW4 4BT, UK
[3]School of Chemistry, University of Glasgow, Joseph Black Building, Glasgow G12 8QQ, UK
[4]Chemical and Biochemical Sciences, Green Process Engineering (CBS), Mohamed VI Polytechnic University, UM6P, Ben Guerir 43150, Morocco
[5]Department of Chemistry, University College London, 20 Gordon Street, London WC1H 0AJ, UK
[6]School of Chemistry, Cardiff University, Park Place, Cardiff CF 10 1AD, UK

CDZ-Y, 0000-0002-8388-1549

In this review, we present the recent progress in ammonia synthesis research using density functional theory (DFT) calculations on various industrial catalysts, metal nitrides and nano-cluster-supported catalysts. The mechanism of ammonia synthesis on the industrial Fe catalyst is generally accepted to be a *dissociative mechanism*. We have recently found, using DFT techniques, that on $Co_3Mo_3N$ (111) surfaces, an *associative mechanism* in the synthesis of ammonia can offer a new low-energy pathway that was previously unknown. In particular, we have shown that metal nitrides that are also known to have high activity for ammonia synthesis can readily form nitrogen vacancies which can activate dinitrogen, thereby promoting the associative mechanism. These fundamental studies suggest that a promising route to the discovery of low-temperature ammonia synthesis catalysts will be to identify systems that proceed via the associative mechanism, which is closer to the nitrogen-fixation mechanism occurring in nitrogenases.

# 1. Introduction

The industrial synthesis of ammonia is reportedly the second largest major-scale chemical synthesis currently performed by mankind [1]. Due to the immense industrial importance of understanding and optimizing this reaction, it is has been the central focus of much research [2–10]. Progress is needed for the production of green ammonia using $H_2$ derived from sustainable sources on a localized scale using catalysts which operate at lower pressures, since currently about 2% of the global annual energy supply is required to produce ammonia industrially [11]. Moreover, it is argued that with the integration of ammonia as a fuel in new applications (e.g. maritime transport), the global need for efficient ammonia production will become even more important [12,13]. Ammonia can be used in internal combustion engines with minor modifications [14] and the first use of liquid anhydrous ammonia as a fuel for motor buses took place in Belgium (1943) [15]. Complete combustion of ammonia produces inert gases according to the following chemical reaction:

$$4NH_3 + 3O_2 \rightarrow 2N_2 + 6H_2O. \tag{1.1}$$

Fixed nitrogen from the Haber–Bosch (H–B) process is also used for the production of biofuels and bioenergy through the use of fertilizers; bioenergy contributes 10% of the global energy requirement, whereas biofuels contribute 1.5% [11].

Nonetheless the most significant current use of ammonia is in the manufacture of fertilizers, which sustains 40% of the global food production and which accounts for the use of 85% of synthetically produced ammonia. The remaining 15% of the ammonia synthesized is used in various industrial applications including the manufacture of polymers, refrigeration fluids and as a reducing agent for emission control in de-$NO_x$ technologies [16].

In figure 1, we present the simplified H–B process used industrially to produce ammonia. The source of hydrogen is usually natural gas but coal and liquid fuels are also used, with sustainably derived hydrogen proving a contemporary focus of attention. These carbon-containing feedstocks can be transformed via steam reforming reactions to hydrogen ($H_2$) and carbon monoxide (CO) (Reaction A). Hydrogen is partially used to remove the oxygen in atmospheric air in order to form a pure feedstream of nitrogen ($N_2$) (Reaction B). Finally, nitrogen and hydrogen are combined over promoted Fe catalysts at high temperatures (400–500°C) and pressures (100–200 atm) in order to produce ammonia (Reaction C). The high temperature is required to make the reaction kinetics fast; however, the reaction is exothermic, which shifts the equilibrium of Reaction C towards the reactants, thereby limiting the conversion. This effect is counterbalanced by the use of high pressure, which shifts the reaction towards the products according to Le Chatelier's principle, suggesting that if the reaction can operate at lower temperatures then the pressure requirement would be reduced while still obtaining reasonable yields.

A significant number of research articles have recently been published reporting the electrochemical reduction of $N_2$ to $NH_3$ which can operate at ambient temperature; however, the discovery of an electrocatalyst with high selectivity remains a challenge [17] although significant progress has been recently made [18]. Skúlason *et al*. have made a thorough computational, density functional theory (DFT) study of various transition-metal electrocatalysts in order to predict the theoretical overpotential required to drive nitrogen reduction [19]. The most active surfaces for the electrocatalytic reduction of $N_2$ were found to be those of Mo, Fe, Rh and Ru.

Computational modelling is now widely used in probing structure and mechanism at the molecular level in catalytic systems. Modelling the ammonia synthesis mechanisms is challenging as there are the possibilities of *associative* (where H or $H_2$ reacts with adsorbed $N_2$) and *dissociative* (where H or $H_2$ reacts with adsorbed N species) pathways [20–22]. The *associative* mechanism could also proceed via an Eley–Rideal/Mars–van Krevelen mechanism where $H_2$ reacts directly with end-on activated $N_2$ at nitrogen vacancies apart from the conventional Langmuir–Hinshelwood mechanism [21,22]. In this review, we present a comparative analysis of the various mechanisms for ammonia synthesis that have been modelled computationally in recent years in order to address the similarities of the various proposed mechanisms and present the main features of the thermo-catalytic and electrocatalytic ammonia synthesis mechanism.

# 2. Systems and mechanisms

## 2.1. Catalysts active for thermochemical ammonia synthesis

The currently used catalyst for the H–B process is an Fe-$K_2O$-$Al_2O_3$ catalyst, although some industrial plants now use a graphite-supported alkali/alkaline earth-promoted Ru catalyst which operates at milder conditions [23].

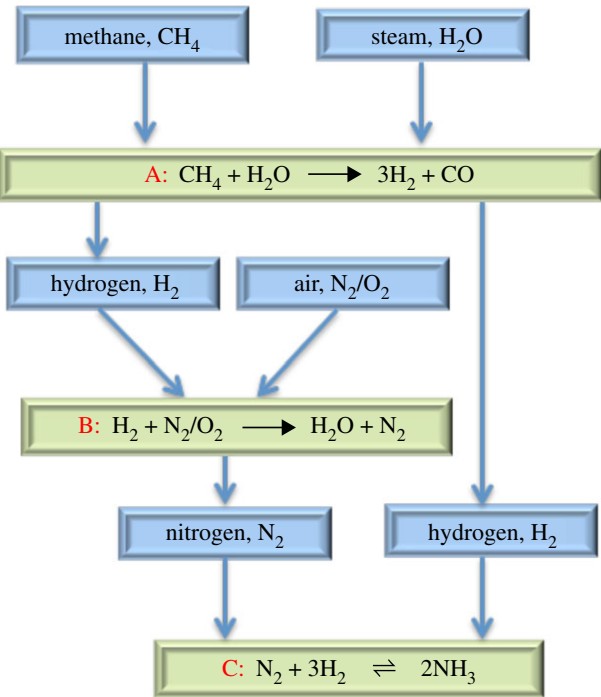

**Figure 1.** Simplified schematic showing some of the basic steps of the industrial H−B process involving steam reforming of methane as the source of $H_2$.

Another catalyst that is active for ammonia synthesis is $Co_3Mo_3N$ based on the pioneering work of Kojima & Aika [24] and Jacobsen [25] especially when it is promoted with a Cs dopant [24,26] but is currently not used industrially.

There are also supported metal ammonia synthesis catalysts that have been recently screened, and in a comparison of a number of systems, $Ru/Al_2O_3$ was observed to have comparatively high activity followed by supported Os and CoRe catalysts [27].

## 2.2. $N_2$ adsorption, activation and dissociation on Fe(111) and Ru(0001)

Several detailed computational studies of the adsorption, activation and dissociation of $N_2$, complementing experiment have been reported in order to determine the activation barrier of this elementary step and to understand the structure of the adsorbates prior to and after the surface reaction. Many of these studies focused on the most reactive surface of iron which is the (111) surface. The adsorption and dissociation of $N_2$ on Fe(111) was studied by Mortensen *et al.* in a detailed DFT study [28] which showed the existence of four different molecularly adsorbed states of $N_2$ of which one is a precursor to the dissociation. The Ru(0001) surface was reported to adsorb $N_2$ in an end-on configuration which rotated to the side-on molecularly adsorbed state, which then dissociated into two HCP adsorption sites [29]. These DFT studies were significant precursor studies to the studies that subsequently modelled the complete reaction mechanism.

## 2.3. Mechanism of ammonia synthesis on Fe and Ru

The generally accepted mechanism for ammonia synthesis on Fe is known from experiment, and it is a Langmuir–Hinshelwood mechanism in which the first step is also the rate-determining step (RDS) [30],

$$N_2(g) + 2^* \rightarrow 2N^*(RDS)$$
$$H_2(g) + 2^* \rightleftharpoons 2H^*$$
$$N^* + H^* \rightleftharpoons NH^* + {}^*$$
$$NH^* + H^* \rightleftharpoons NH_2^* + {}^*$$
$$NH_2^* + H^* \rightleftharpoons NH_3^* + {}^*$$
and
$$NH_3^* \rightleftharpoons NH_3(g) + {}^*$$

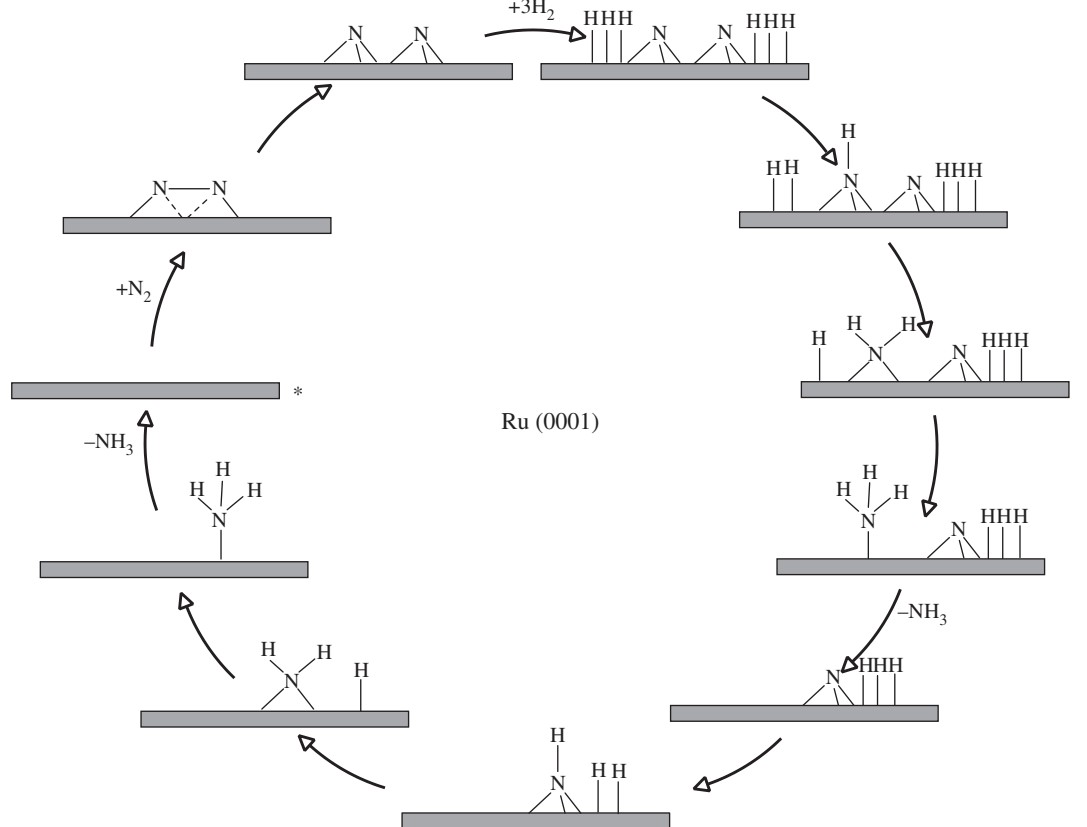

**Figure 2.** Simplified schematic of *dissociative* mechanism for ammonia synthesis on Ru(0001) surfaces according to the work of Logadóttir & Nørskov [31]. Reaction starts at the asterisk.

where the lone * represents a vacant surface site. Notice that all reaction steps are in equilibrium apart from the dinitrogen dissociation step. In this mechanism, dissociation of the triple bond in dinitrogen is very energy intensive and therefore the high-pressure/temperature requirements of the currently used industrial catalyst. This mechanism is a dissociative mechanism and is the one accepted for the two currently used industrial catalyst Fe and Ru.

## 2.4. Ammonia synthesis mechanism on Ru

Ruthenium is a more active catalyst for ammonia synthesis than iron, [31] but ruthenium is more expensive, and it has a shorter lifetime than the Fe catalyst [32]. A non-dissociative mechanism for ammonia synthesis instead of the conventional dissociative pathway at a stepped site on lithium-doped ruthenium nanoparticles was modelled by Zheng *et al.* [33]. Logadóttir & Nørskov [31] investigated via DFT calculations, the mechanism of ammonia synthesis (shown in figure 2) on flat, and stepped ruthenium (0001) surfaces. They find that, as with the iron catalyst, the RDS is the dissociation of $N_2$, which was calculated to have a lower barrier at steps. They modelled a mechanism that is the same as the one occurring on iron. Furthermore, this computational study suggested that the promotion of reactants by alkali metals takes place by stabilizing the transition state of $N_2$ dissociation and by destabilizing the *NH species on the surface of the catalyst. The dissociative mechanism that was modelled (figure 2) is that of a typical Langmuir–Hinshelwood mechanism in which $N_2$ adsorbs dissociatively followed by the dissociation of three $H_2$ forming *N and *H species, respectively. These species are co-adsorbed on the surface of the catalyst in positions that allow the reaction of *H with *N. In the next step, hydrogen atoms are added to nitrogen atoms forming first *NH, then *$NH_2$ and finally *$NH_3$ which desorbs from the catalytic surface. After the desorption of the first $NH_3$, the remaining *N is hydrogenated via a similar process to the previous step, in order to form a second $NH_3$ molecule. The intermediates *N and *H both are adsorbed at FCC sites, which is in agreement with low-energy electron diffraction study coupled to DFT [34]. *NH adsorbs at an HCP site and reacts with additional H to form *$NH_2$ which moves to a bridge site. The ammonia that is

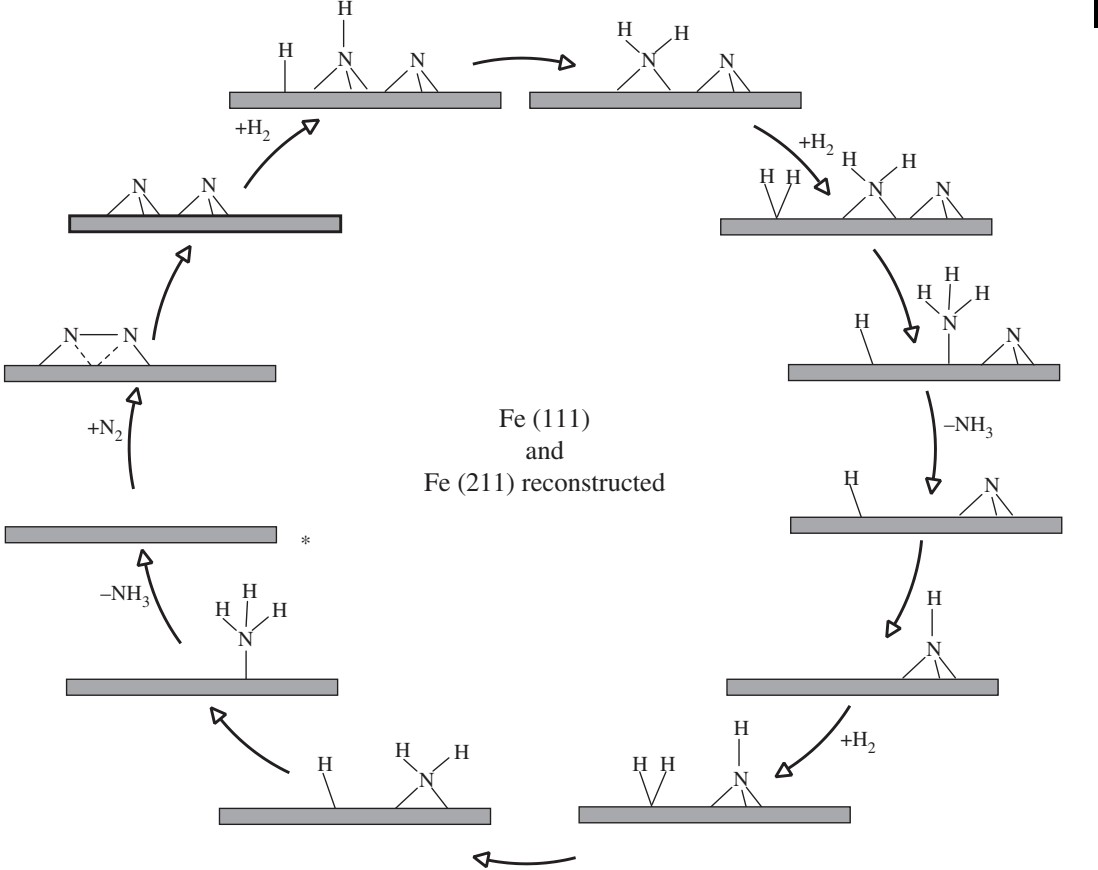

**Figure 3.** Simplified schematic of the *dissociative* mechanism for ammonia synthesis on Fe(111) [36] and Fe(211) [37] reconstructed surfaces according to the work Qian *et al.* and Fuller *et al.*, respectively. Reaction starts at the asterisk.

formed is adsorbed to an on-top site through its lone electron pair. The barriers for the hydrogenation steps have been found to be similar. In particular, the first hydrogenation step has a barrier of 1.2 eV molecule$^{-1}$, increasing by 0.1 eV per hydrogen atom added [31]; these results are in good agreement with those of Zhang *et al.* [35].

## 2.5. Ammonia synthesis mechanism on Fe(111) and Fe(211) reconstructed

The ammonia synthesis mechanism on Fe(111) [36] and Fe(211) [37] was modelled by Qian *et al.* and Fuller *et al.*, respectively, and the initial step was found to be side-on adsorption of dinitrogen that dissociates through a high barrier elementary reaction step, forming adsorbed nitrogen at threefold hollow iron sites. The dissociation of $N_2$ in the first step of the reaction mechanism indicates that it is again a *dissociative* mechanism in agreement with the mechanism known for Fe(111) and other iron surfaces [30]. Hydrogen dissociatively adsorbs forming adsorbed atomic hydrogen *H and >NH species. These >NH species undergo further hydrogenation with atomic hydrogen forming first $NH_2$ species and then $NH_3$, which desorbs from the surface of the catalyst. The second hollow-bound nitrogen reacts with the surface hydrogen atoms from >NH. A third hydrogen molecule dissociatively adsorbs upon the catalyst surface and starts hydrogenating >NH, forming first >$NH_2$ that is bridge-bound and subsequently adsorbed -$NH_3$, which then desorbs from the surface regenerating the catalytic site of Fe(111) and reconstructed Fe(211). A summary of the mechanism is illustrated in figure 3.

The authors of the investigations of the ammonia synthesis mechanism on Fe(111) [36] and Fe(211) [37] compared their calculated barriers to the experimentally determined barrier. In particular, Strongin *et al.* [38] report for the temperature-programmed desorption of ammonia from Fe(111) a desorption barrier of 0.82 eV, which Qian *et al.* [36] calculate to be between 0.89 and 1.00 eV. Furthermore, the calculated barrier for $NH_x$ dexydrogenation (*$NH_x \rightarrow$ *$NH_{x-1}$+*H) ranges from 0.53 to 0.63 eV; the barrier for $NH_3$ desorption ranges from 0.76 to 0.87 eV; the barrier of $N_2$ dissociation is calculated as 1.27 eV and the barrier for N-adatom recombination is 2.04 eV [36]. The last value,

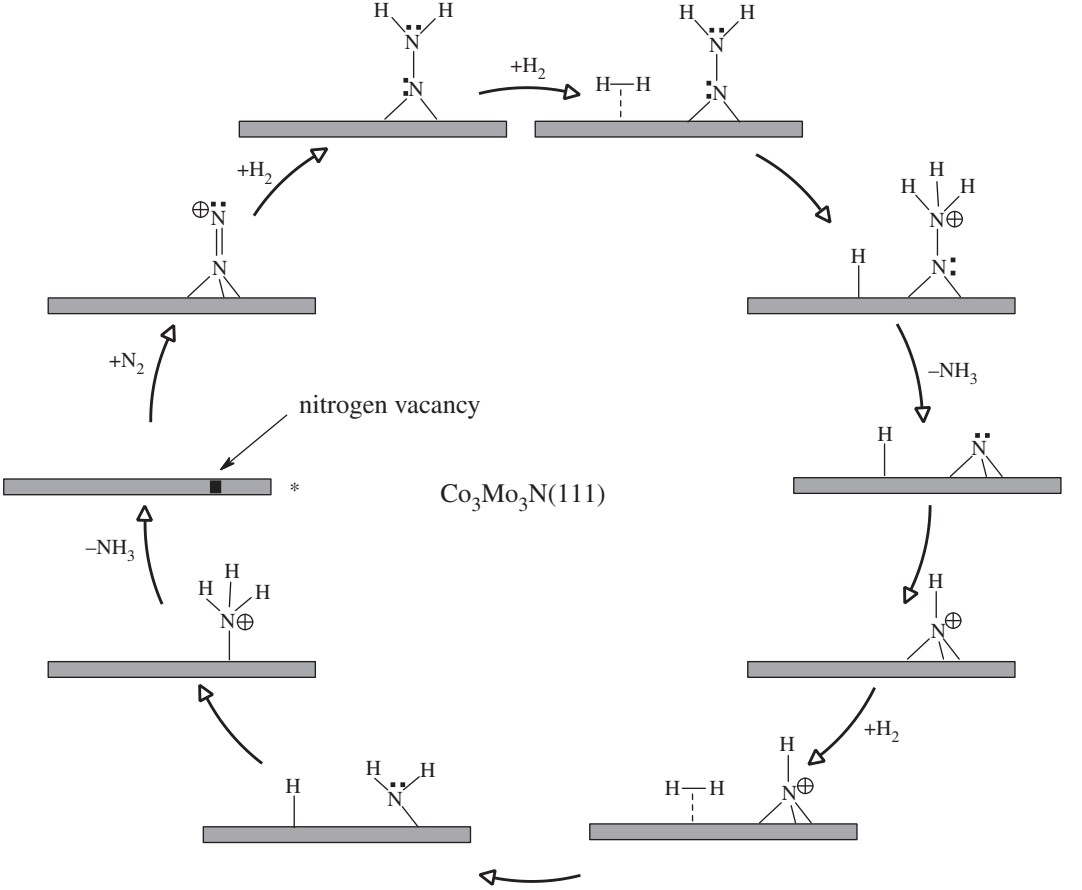

**Figure 4.** Simplified schematic of *associative* Eley–Rideal/Mars–van Krevelen mechanism for ammonia synthesis on Co₃Mo₃N according to Zeinalipour-Yazdi *et al.* [22]. Reaction starts at the asterisk.

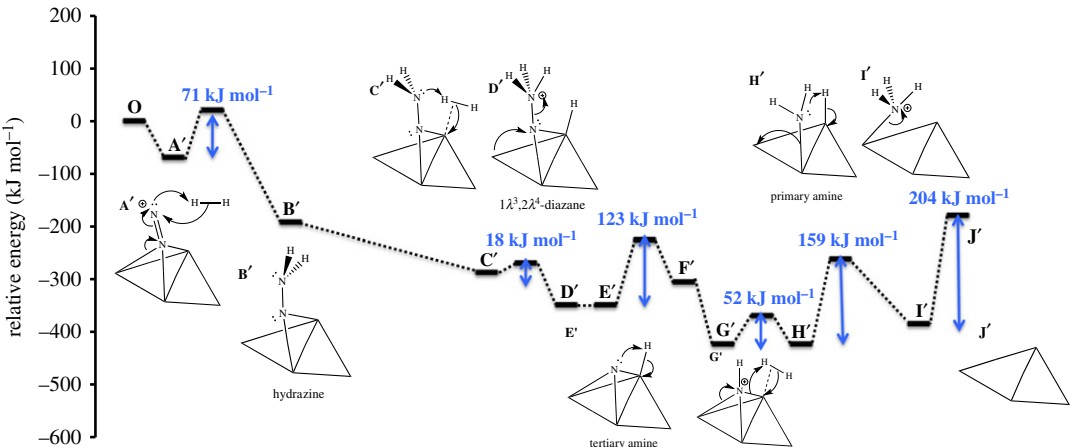

**Figure 5.** Potential energy diagram of the associative Eley–Rideal/Mars–van Krevelen mechanism for ammonia synthesis on Co₃Mo₃N proposed by Zeinalipour-Yazdi *et al.* [22].

which is 47 kcal mol⁻¹ (2.04 eV), compares very well with the experimental value that Ertl *et al.* [39,40] determined on polycrystalline Fe wires, which was found to be 45 ± 5 kcal mol⁻¹.

## 2.6. Ammonia synthesis mechanism on Co₃Mo₃N

The present authors have investigated in detail two mechanisms [21,22] of ammonia synthesis on Co₃Mo₃N-(111) surfaces for which the lower energy pathway is depicted in figure 4, and the potential energy diagram is given in figure 5. It is an *associative* Eley–Rideal/Mars–van Krevelen mechanism which happens at surface

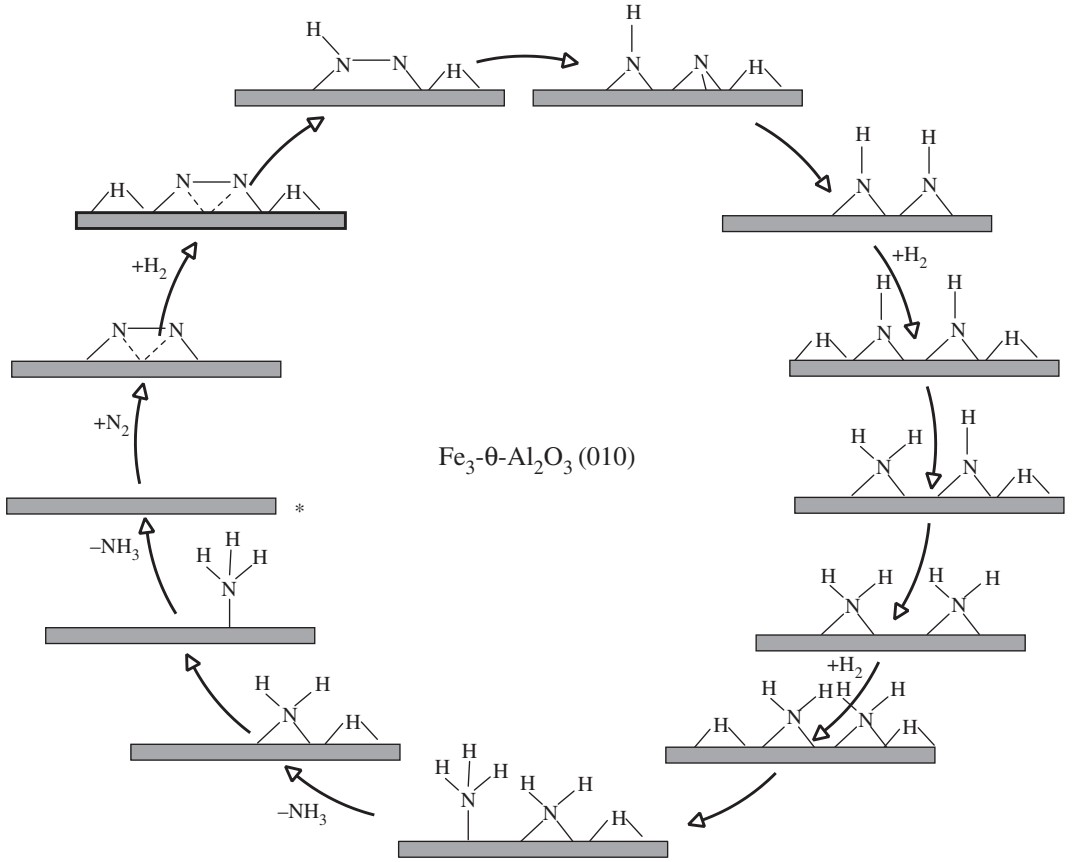

**Figure 6.** Simplified schematic of *associative* mechanism for ammonia synthesis on $Fe_3$-$\theta$-$Al_2O_3$-supported clusters according to the mechanism proposed by Liu *et al.* [43]. Reaction starts at the asterisk.

nitrogen vacancies of threefold bound nitrogen. Gas phase nitrogen chemisorbs in an end-on configuration, which then further reacts with hydrogen, coming directly from the gas phase. The direct reaction of pre-activated $N_2$ had not previously been considered in the mechanism of ammonia synthesis for metal nitrides. Furthermore, in this associative mechanism, the activated $N_2$ first becomes hydrogenated, which weakens considerably the strong N–N triple bond. The hydrogenation forms >$NNH_2$, which is an intermediate also found in the biological nitrogen-fixation mechanism (>$NNH_2$, similar to Mo=$NNH_2$ [41] and Fe–$NNH_2$ [42]), which explains the high activity of $Co_3Mo_3N$ for ammonia synthesis, even at lower temperatures. After the first hydrogenation step, which has a low barrier (71 kJ mol$^{-1}$), a second molecular hydrogen can adsorb at the nearby position on the $Mo_3N$ framework. The molecular hydrogen dissociates reacting with >$NNH_2$ and forming >$NNH_3$ and atomic H species. The diazane-like intermediate decomposes readily in a low-energy barrier process (18 kJ mol$^{-1}$) to form $NH_3$(g) and atomic N which remains adsorbed to the threefold hollow of the $Mo_3$ site. The remaining atomic H at the active site reacts with atomic N forming >NH, in a moderate barrier process of 123 kJ mol$^{-1}$. Additional $H_2$ adsorbs to the active site of the catalyst forming a bridged-bound-like intermediate –$NH_2$, in a low-barrier step of 52 kJ mol$^{-1}$. –$NH_2$ reacts with atomic H species in a 159 kJ mol$^{-1}$ barrier process, to form surface-adsorbed ammonia, –$NH_3$. This surface-adsorbed ammonia is very strongly bound to the active site of the catalyst making its strongly edothermic desorption energy of 204 kJ mol$^{-1}$, the RDS of the mechanism of ammonia synthesis on $Co_3Mo_3N$(111) surfaces [21,22].

## 2.7. Ammonia synthesis mechanism on a $\theta$-$Al_2O_3$-supported $Fe_3$ catalyst

There are also DFT studies of the mechanism of ammonia synthesis on supported metal clusters [43]. These have also looked at an associative mechanism. These calculations show that dinitrogen is activated in a side-on configuration on three Fe atoms, each iron forming a bridged bond with the nitrogen atoms (figure 6). Due to the highly uncoordinated structure of the supported $Fe_3$ cluster, $H_2$ adsorbs dissociatively at two bridged >H adsorption sites. The activated >NN< forms a chemical bond with one of the bridged >H, forming >NNH<, which subsequently dissociates to adsorbed atomic nitrogen >N and >NH. The authors investigated other

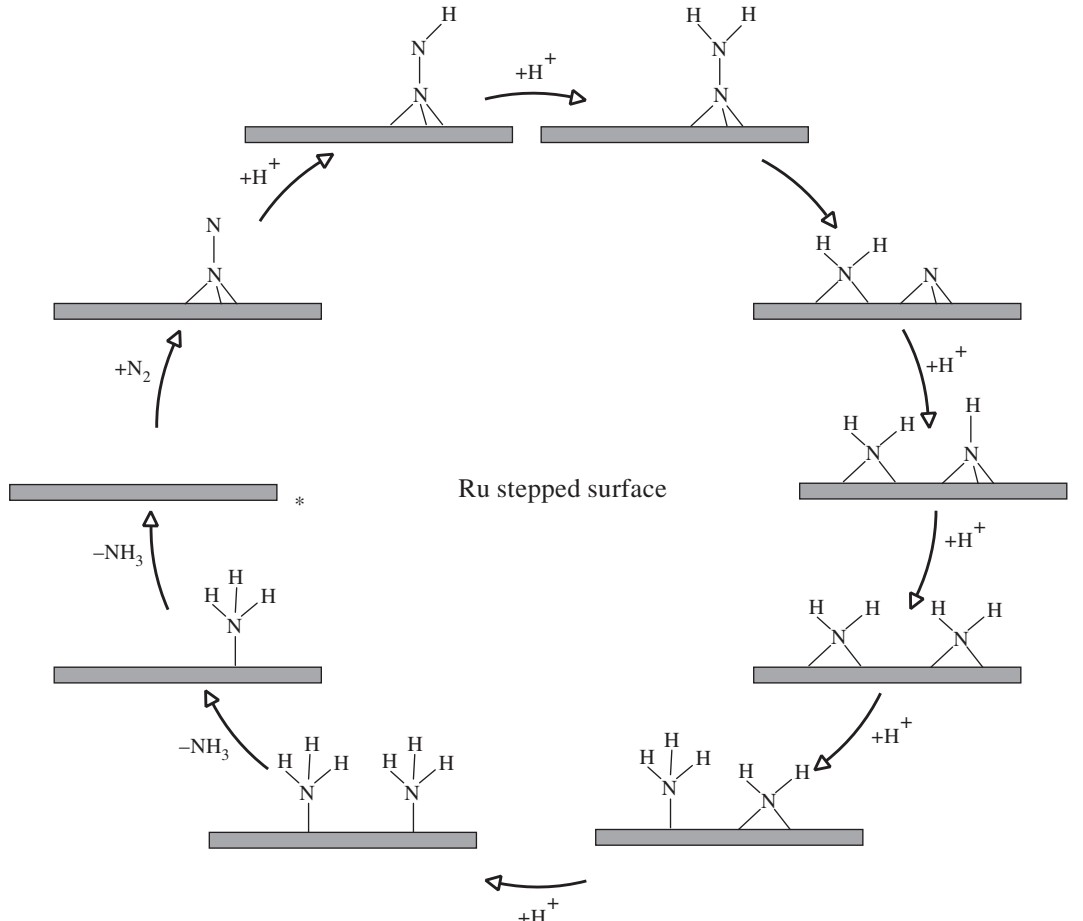

**Figure 7.** Simplified schematic of *associative* mechanism for ammonia synthesis on Ru stepped surface according to the mechanism proposed by Back & Jung [48]. Reaction starts at the asterisk.

hydrogenation pathways that lead to the formation of >HNNH< and >NNH$_2$, but they indicate that these intermediates may be important in homogeneous catalysis and the enzymatic mechanism [44–47]. In the following step, the second bridged >H reacts with the adsorbed >N, forming a second >NH intermediate. After this step, another H$_2$ dissociatively adsorbs to the Fe$_3$ cluster forming two more bridged >H species. These bridged >H species react with >NH forming two bridged adsorbed >NH$_2$ species. Subsequently, another H$_2$ reacts with the cluster forming two adsorbed bridged >H species. These >H species react with the >NH$_2$ species forming first adsorbed −NH$_3$, which desorbs followed by the hydrogenation on the second >NH$_2$, resulting in a second −NH$_3$. This NH$_3$ desorbs from the surface freeing the catalytic site of the Fe$_3$-θ-Al$_2$O$_3$ (010) catalyst.

The adsorption and activation of dinitrogen on the Fe$_3$ clusters is exothermic by −0.75 eV. The first barrier for hydrogenation is 0.98 eV forming >NNH. The >NNH< intermediate migrates from the Fe$_3$ cluster to the Fe$_3$-θ-Al$_2$O$_3$ interface with a barrier of 0.45 eV in a structure in which N is coordinated by two Fe atoms and >NH by a Fe atom and an Al ion. The >NNH< intermediate dissociates to >NH and threefold bound nitrogen in a 0.45 eV barrier process. The second hydrogenation step has a barrier of 1.11 eV forming two >NH species, which is followed by H$_2$ adsorption which is exothermic by −0.61 eV. The barriers for conversion of >NH to >NH$_2$ are 1.32 and 1.22 eV, which is followed by another exothermic adsorption of H$_2$, which is −1.02 eV. The barrier for the step going from >NH$_2$ to −NH$_3$ is 0.61 eV, followed by an endothermic step for ammonia desorption of 1.01 eV. The last hydrogenation step has a barrier of 1.32 eV followed by endothermic desorption of NH$_3$ which is 0.99 eV.

## 2.8. Mechanism of the electrochemical synthesis of ammonia on stepped Ru

Back & Jung [48] showed that on Ru-stepped surfaces, the electrochemical mechanism for ammonia synthesis follows an associative reaction mechanism. In this mechanism, dinitrogen becomes activated

in an end-on configuration which receives hydrogen proton from the solution forming >NNH. In their mechanism, they also considered elementary steps that undergo dissociation of the N–N bond at different stages of hydrogenation for species such as >NNH$_2$, >HNNH<, >NH$_2$NH< and >NH$_2$NH$_2$< but found that the lowest energy pathway is when >NNH$_2$ dissociates to >NH$_2$ and threefold hollow adsorbed *N. The >NH$_2$ species and *N obtain an H$^+$ from the solution, forming bridged >NH$_2$ and threefold hollow-bound >NH. In a subsequent hydrogenation step, >NH$_2$ reacts with H$^+$ forming on-top bound −NH$_3$. This elementary reaction repeats for the second >NH$_2$ forming two −NH$_3$ species on the step of the Ru catalyst. These two −NH$_3$ species later desorb from the surface of the catalyst in two consecutive steps regenerating the active site at the step of Ru. Figure 7 provides a diagrammatic summary of the mechanism.

The energetic barriers for these elementary reaction steps were also calculated. In particular, they find that the dissociation barrier of activated N$_2$ to atomically adsorbed *N is 1.69 eV. The corresponding barrier for >NNH< dissociation to *N and >NH was found to be lower, 1.45 eV. The dissociation barrier for >HNNH< to two >NH was found to be even lower, 1.36 eV. Interestingly, the lowest barrier was found to be the dissociation of >NNH$_2$< to atomically adsorbed *N and >NH$_2$ which was only 0.20 eV. A barrier of 0.52 eV was found for the dissociation of >NHNH$_2$< to >NH and >NH$_2$. Finally, a very low barrier was found for the fully hydrogenated species >NH$_2$NH$_2$<, which decomposes to two bridged-bound >NH$_2$ at the Ru step. Interestingly the elementary step in which the dissociation of >NNH$_2$< happens is the lowest energy pathway on the potential energy diagram of the mechanism, which indicates that the mechanism is an associative mechanism in which the >NNH$_2$< intermediate dissociates to form >NH$_2$ and threefold bound N species.

## 3. Conclusion

Examination of the DFT calculated reaction mechanisms of various pathways in the thermochemical and electrochemical ammonia synthesis has certainly enhanced our understanding of the elementary reaction steps. There are several similar features that we find by comparative analysis of this mechanism, which could become future directions of computational and experimental studies of the ammonia synthesis reaction mechanism:

1. Consideration of associative mechanisms of ammonia synthesis in certain systems can lead to the suggested occurrence of new lower energy pathways for ammonia synthesis that are closer to the nitrogen-fixation mechanisms occurring in nitrogenases.
2. There are certain common intermediates such as the threefold hollow-bound *N, the bridged bond >NH$_2$ and on-top bond −NH$_3$ that can be found in the mechanisms investigated on Fe, Ru and Co$_3$Mo$_3$N surfaces and clusters.
3. Certain new intermediates such as >NNH$_2$ and >NNH$_3$ are found in the associative mechanism for ammonia synthesis on Co$_3$Mo$_3$N which need to be confirmed spectroscopically.
4. The barrier for NH$_3$ formation from >NNH$_3$ is very low compared with the barrier of NH$_3$ desorption from Co$_3$Mo$_3$N.
5. When the mechanism follows an associative pathway, the RDS may become the desorption of NH$_3$ from the catalyst active site.

Overall, although it is generally accepted that the mechanism of ammonia synthesis on the two main industrial catalysts Fe and Ru is a dissociative mechanism, there is increasing evidence from DFT calculations that on systems such as Co$_3$Mo$_3$N and stepped Ru surfaces as well as on small clusters of Fe$_3$ on alumina, in the thermochemical catalytic cycle of ammonia synthesis and the mechanism of electrochemical synthesis of ammonia on Ru surfaces, the mechanism follows an associative pathway. This pathway is generally operative in the enzymatic catalysts, and it would be interesting to see whether there are heterogeneous catalytic systems that follow this mechanism that may produce ammonia at lower temperatures and pressures.

Data accessibility. This article has no additional data.

Authors' contributions. C.D.Z.-Y. planned and carried out most of the simulations and drafted the manuscript. J.S.J.H. edited and revised the manuscript and formulated some of the ideas. S.L. edited and revised the manuscript and formulated some of the ideas. C.R.A.C. conceived the research topic, supervised the project, edited and revised the manuscript.

Competing interests. We declare we have no competing interests.

Funding. The authors acknowledge financial support from EPSRC (grant nos. EP/L026317/1 and EP/L02537X/1). The authors acknowledge EPSRC-funded computer resources (grant no. EP/L000202/1) at ARCHER UK National

Supercomputing Service (http://www.archer.ac.uk) received via our membership of the UK's HPC Materials Chemistry Consortium.

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
