## [Peer Review File · Royal Society Open Science]

Review History

RSOS-210952.R0 (Original submission)

Review form: Reviewer 1

Is the manuscript scientifically sound in its present form?

No

Are the interpretations and conclusions justified by the results?

No

Is the language acceptable?

Yes

Do you have any ethical concerns with this paper?

No

Have you any concerns about statistical analyses in this paper?

No

Recommendation?

Reject

Comments to the Author(s)

In this paper, the authors reviewed the reaction mechanism of ammonia synthesis and summarized it into two categories: dissociative mechanism and associative mechanism. The authors also discussed the reaction mechanism for electrochemical ammonia synthesis. As a review paper, I believe this paper is poorly written and offers little insight into the reaction mechanisms. The paper simply rephrases the statements reported in the literature and what is worse, makes these statements even difficult to understand. For example, the authors used five "simplified schematics" to illustrate the dissociative and associative reaction mechanisms on different catalysts. These figures are very confusing and less informative. In the main text, the authors thoroughly describe these reactions on different sites, which suggests that the reaction sites are of critical importance to the reaction and largely determine the possible reaction mechanism. Nevertheless, such important information is not indicated in the figures.

The authors cited too many old works, though ammonia synthesis is an ancient topic. It is better the authors briefly summarize the old findings and present the recent three- or five-years research progress.

For the electrochemical ammonia synthesis, the authors only summarized the findings of one paper (ref 48) to demonstrate the conclusion. This is a rather weak way. It is also clear that the barrier calculations in ref 48 are completely wrong. Neither the authors nor the authors in ref 48 realize the difference between thermochemical and electrochemical reactions.

Review form: Reviewer 2**Is the manuscript scientifically sound in its present form?**

Yes

Are the interpretations and conclusions justified by the results?

Yes

Is the language acceptable?

Yes

Do you have any ethical concerns with this paper?

No

Have you any concerns about statistical analyses in this paper?

No

Recommendation?

Accept with minor revision (please list in comments)

Comments to the Author(s)

This is an excellent review of the literature associated with the theoretical studies for ammonia synthesis using DFT methods. The paper is well written and covers the literature in detail. Some comparison is made between the studies is provided but this should be expanded and it would

be useful to also provide a more detailed perspective of future directions and catalyst design possibilities if possible. Overall, this paper is publishable with minor corrections

Decision letter (RSOS-210952.R0)

Dear Dr Zeinalipour-Yazdi:

Title: A comparative analysis of the mechanisms of ammonia synthesis on various catalysts using DFT

Manuscript ID: RSOS-210952

The editor assigned to your manuscript has now received comments from reviewers. We would like you to revise your paper in accordance with the referee and Subject Editor suggestions which can be found below (not including confidential reports to the Editor). Please note this decision does not guarantee eventual acceptance.

Please submit your revised paper before 02-Sep-2021. Please note that the revision deadline will expire at 00.00am on this date. If we do not hear from you within this time then it will be assumed that the paper has been withdrawn. In exceptional circumstances, extensions may be possible if agreed with the Editorial Office in advance. We do not allow multiple rounds of revision so we urge you to make every effort to fully address all of the comments at this stage. If deemed necessary by the Editors, your manuscript will be sent back to one or more of the original reviewers for assessment. If the original reviewers are not available we may invite new reviewers.

On behalf of the Subject Editor Professor Anthony Stace and the Associate Editor Professor Kim Jelfs.

RSC Associate Editor:
Comments to the Author:
(There are no comments.)

RSC Subject Editor:
Comments to the Author:
(There are no comments.)

Reviewers' Comments to Author:

Reviewer: 1

Comments to the Author(s)

In this paper, the authors reviewed the reaction mechanism of ammonia synthesis and summarized it into two categories: dissociative mechanism and associative mechanism. The authors also discussed the reaction mechanism for electrochemical ammonia synthesis. As a review paper, I believe this paper is poorly written and offers little insight into the reaction mechanisms. The paper simply rephrases the statements reported in the literature and what is worse, makes these statements even difficult to understand. For example, the authors used five "simplified schematics" to illustrate the dissociative and associative reaction mechanisms on different catalysts. These figures are very confusing and less informative. In the main text, the authors thoroughly describe these reactions on different sites, which suggests that the reaction sites are of critical importance to the reaction and largely determine the possible reaction mechanism. Nevertheless, such important information is not indicated in the figures.

The authors cited too many old works, though ammonia synthesis is an ancient topic. It is better the authors briefly summarize the old findings and present the recent three- or five-years research progress.

For the electrochemical ammonia synthesis, the authors only summarized the findings of one paper (ref 48) to demonstrate the conclusion. This is a rather weak way. It is also clear that the barrier calculations in ref 48 are completely wrong. Neither the authors nor the authors in ref 48 realize the difference between thermochemical and electrochemical reactions.

Reviewer: 2

Comments to the Author(s)

This is an excellent review of the literature associated with the theoretical studies for ammonia synthesis using DFT methods. The paper is well written and covers the literature in detail. Some comparison is made between the studies is provided but this should be expanded and it would be useful to also provide a more detailed perspective of future directions and catalyst design possibilities if possible. Overall, this paper is publishable with minor corrections

Author's Response to Decision Letter for (RSOS-210952.R0)

See Appendix A.

RSOS-210952.R1 (Revision)

Review form: Reviewer 1

Is the manuscript scientifically sound in its present form?

Yes

Are the interpretations and conclusions justified by the results?

Yes

Is the language acceptable?

Yes

Do you have any ethical concerns with this paper?

No

Have you any concerns about statistical analyses in this paper?

No

Recommendation?

Accept as is

Comments to the Author(s)

I think the revised manuscript is good for publication.

Decision letter (RSOS-210952.R1)

Dear Dr Zeinalipour-Yazdi:

Title: A comparative analysis of the mechanisms of ammonia synthesis on various catalysts using DFT

Manuscript ID: RSOS-210952.R1

It is a pleasure to accept your manuscript in its current form for publication in Royal Society Open Science. The chemistry content of Royal Society Open Science is published in collaboration with the Royal Society of Chemistry.

Yours sincerely,
Dr Ellis Wilde
Publishing Editor, Journals

On behalf of the Subject Editor Professor Anthony Stace and the Associate Editor Professor Kim Jelfs.

RSC Associate Editor
Comments to the Author:
(There are no comments.)

RSC Subject Editor
Comments to the Author:
(There are no comments.)

Reviewer(s)' Comments to Author:
Reviewer: 1

Comments to the Author(s)
I think the revised manuscript is good for publication.

Appendix A

Reviewer: 1

Comments to the Author(s)

In this paper, the authors reviewed the reaction mechanism of ammonia synthesis and summarized it into two categories: dissociative mechanism and associative mechanism. The authors also discussed the reaction mechanism for electrochemical ammonia synthesis. As a review paper, I believe this paper is poorly written and offers little insight into the reaction mechanisms. The paper simply rephrases the statements reported in the literature and what is worse, makes these statements even difficult to understand. For example, the authors used five "simplified schematics" to illustrate the dissociative and associative reaction mechanisms on different catalysts. These figures are very confusing and less informative. In the main text, the authors thoroughly describe these reactions on different sites, which suggests that the reaction sites are of critical importance to the reaction and largely determine the possible reaction mechanism. Nevertheless, such important information is not indicated in the figures.

>>>Response to Reviewer: We thank the Reviewer for the constructive criticism. The reasons we used simplified schematics and not explicit schematics of the various reaction mechanisms is that we wanted to provide a complete overview of the various mechanisms occurring on different systems in a systematic way in which the reader can compare to one another the various mechanisms. We believe that this approach will be useful for readers not directly familiar with the topic of ammonia synthesis mechanisms to obtain a general overview of the various mechanisms occurring and explore some of the basic differences among the mechanisms. Furthermore, this comparative analysis of the various mechanisms for ammonia synthesis that were modelled computationally in recent years offers the similarities of the various proposed mechanisms and present the main features of the thermo-catalytic and electrocatalytic ammonia synthesis mechanism. We have made changes to the manuscript to improve our statements about the various mechanism and make these statements more reader friendly. The changes are highlighted in the revised manuscript. On page 6 of the manuscript we have added the sentence " These species are co-adsorbed on the surface of the catalyst in positions that allow the reaction of *H with *N." On page 8 we have added the sentence " forming first NH₂ species and then NH₃ which desorbs from the surface of the catalyst". On page 8 we have deleted the sentence "Following this, a further H₂ molecule adsorbs on the surface of iron dissociatively. The adsorbed hydrogen species then react with >NH₂ forming on-top adsorbed ammonia -NH₃ which desorbs from the catalyst". On page 10 we have added the words "strongly endothermic". On page 14 we have added the sentence " to form >NH₂ and three-fold bound N species"

The authors cited too many old works, though ammonia synthesis is an ancient topic. It is better the authors briefly summarize the old findings and present the recent three- or five-years research progress.

>>> Response to Reviewer: The Reviewer correctly indicates that we have cited many old papers about ammonia synthesis. We believe that this was necessary to give the appropriate credit to the relevant computational studies. We only mention papers that have published a mechanism for ammonia synthesis that show mechanistic insight about the reaction mechanism. An important step in the mechanism of ammonia synthesis is the activation of dinitrogen therefore we have also included papers that just study the adsorption and activation of dinitrogen.

For the electrochemical ammonia synthesis, the authors only summarized the findings of one paper (ref 48) to demonstrate the conclusion. This is a rather weak way. It is also clear that the barrier calculations in ref 48 are completely wrong. Neither the authors nor the authors in ref 48 realize the difference between thermochemical and electrochemical reactions.

>>> Response to Reviewer: We acknowledge that some information given in the published computational studies maybe incomplete or incorrect. We have only mention this study from a historical perspective as it was the first study of the mechanism of the electrochemical reaction mechanism. If there are other papers that should have been included that mention the complete mechanism of ammonia synthesis electrochemically and are correct we ask the reviewer to provide us with these references so they can be included.

Reviewer: 2

Comments to the Author(s)

This is an excellent review of the literature associated with the theoretical studies for ammonia synthesis using DFT methods. The paper is well written and covers the literature in detail. Some comparison is made between the studies is provided but this should be expanded and it would be useful to also provide a more detailed perspective of future directions and catalyst design possibilities if possible. Overall, this paper is publishable with minor corrections

>>> Response to Reviewer: We thank the Reviewer for the positive recommendation. We have already published a paper about the mechanism of ammonia synthesis which provides future directions (The integration of experiment and computational modelling in heterogeneously catalysed ammonia synthesis over metal nitrides, CD Zeinalipour-Yazdi, JSJ Hargreaves, S Laassiri, CRA Catlow, Physical Chemistry Chemical Physics 20 (34), 21803-21808). In this review we mention some new aspects of the ammonia synthesis mechanism that are worth of further consideration for future computational and experimental studies. For example the spectroscopic identification of certain reactions intermediates and the consideration of a potential low temperature associative mechanism which bares features of the enzymatic reaction pathway. We have added a sentence on page 14 of the manuscript that that indicates that the findings of the comparative analysis of the ammonia synthesis reaction mechanism could become future direction for computational and experimental studies. The sentence added is " which could become future directions of

computational and experimental studies of the ammonia synthesis reaction mechanism"